# C-edge loops of arrestin function as a membrane anchor

Ciara C.M. Lally[1], Brian Bauer[1], Jana Selent[2] & Martha E. Sommer[1]

G-protein-coupled receptors are membrane proteins that are regulated by a small family of arrestin proteins. During formation of the arrestin–receptor complex, arrestin first interacts with the phosphorylated receptor C terminus in a pre-complex, which activates arrestin for tight receptor binding. Currently, little is known about the structure of the pre-complex and its transition to a high-affinity complex. Here we present molecular dynamics simulations and site-directed fluorescence experiments on arrestin-1 interactions with rhodopsin, showing that loops within the C-edge of arrestin function as a membrane anchor. Activation of arrestin by receptor-attached phosphates is necessary for C-edge engagement of the membrane, and we show that these interactions are distinct in the pre-complex and high-affinity complex in regard to their conformation and orientation. Our results expand current knowledge of C-edge structure and further illuminate the conformational transitions that occur in arrestin along the pathway to tight receptor binding.

[1] Institute of Medical Physics and Biophysics (CC2), Charité Medical University, Charitéplatz 1, Berlin 10117, Germany. [2] Research Programme on Biomedical Informatics, Department of Experimental and Health Sciences, Pompeu Fabra University, Hospital del Mar Medical Research Institute, Carrer del Dr. Aiguader, 88, Barcelona 08003, Spain. Correspondence and requests for materials should be addressed to J.S. (email: jana.selent@upf.edu) or to M.E.S. (email: martha.sommer@charite.de).

G-protein-coupled receptors (GPCRs) comprise a large and diverse family of membrane proteins in animals. GPCRs mediate signal transduction in nearly all sensory and physiological systems and bind a wide range of ligands including small molecules, peptides and proteins. Binding of agonist stabilizes an active conformation of the seven transmembrane helical bundle of the receptor in which the cytoplasmic face is open to bind G protein[1]. The G protein is thereby activated and mediates further cell signalling. The active receptor is also phosphorylated on multiple sites on its carboxy (C) terminus or cytoplasmic loops by GPCR kinases (GRK)[2], which facilitates binding of the protein arrestin[3].

Remarkably, all GPCRs are regulated by only four different arrestins. Arrestin-1 and -4 are expressed in photoreceptor cells in the retina and interact with the visual opsins, and arrestin-2 and -3 (also called β-arrestin 1 and 2, respectively) are expressed ubiquitously and interact with hundreds of different GPCRs[4]. Arrestin binding deactivates receptor signalling by blocking G-protein binding, and the β-arrestins additionally mediate receptor endocytosis and trafficking by recruiting elements of the cellular internalization machinery like clathrin and AP2 to the receptor[5]. The β-arrestins interact with hundreds of other proteins with a wide array of functions, including signalling kinases and phosphatases, ubiquitin ligases, transcription factors, cytoskeletal elements and many more[6]. The β-arrestins mediate their own signalling networks[7,8].

Arrestins are composed of near-symmetric amino (N)- and C-domains, which resemble two clamshells placed end-to-end (Fig. 1a,b). A long C-terminal tail (C-tail) interacts extensively with the N-domain and stabilizes the basal conformation. Arrestins are activated for receptor binding by the phosphorylated C terminus of the receptor, and this initial low-affinity interaction is termed the pre-complex (Fig. 1c). The phosphorylated receptor C terminus displaces the arrestin C-tail, which results in an ~21° rotation of the N- and C-domains against each other and an increase in flexibility in receptor-binding loops in the central crest region (see Fig. 1a)[9–11]. These conformational changes facilitate tight binding and transition to the high-affinity complex (Fig. 1c). Arrestins can also be activated for receptor binding by C-tail truncation, such as in the naturally occurring splice variant of arrestin-1 called p44 (Fig. 1b), which lacks the C-tail and thus exists in a pre-active form that can bind active receptor independent of receptor phosphorylation[12,13].

The first crystal structure of arrestin in complex with an active GPCR was recently published[11]. A stable complex was achieved by introducing activating mutations into both binding partners, mouse arrestin-1 and human opsin, and fusing the N terminus of arrestin to the C terminus of the receptor via a flexible linker. Despite these modifications and the lack of receptor phosphorylation, this structure indicates how the conformational changes associated with arrestin activation facilitate coupling to the active receptor. Notably, the orientation of arrestin in the crystal structure of the Ops* arrestin-1 fusion complex suggests that the 344-loop (residues S336–S344 in bovine arrestin-1) within the distal edge of the C-domain (C-edge, see Fig. 1a) would interact with the membrane adjacent to the receptor[11]. Unfortunately, this loop is not completely resolved, and no membrane is present. Before the crystal structure, the interaction of the 344-loop with the membrane was proposed by our group based on site-directed fluorescence experiments[14].

The current study explores the role of the 344-loop and the C-edge in membrane anchoring of arrestin. Our experimental approach combines molecular dynamics simulations (in silico) and site-directed fluorescence spectroscopy (in vitro). For in vitro experiments, we used bovine arrestin-1 and rhodopsin, which is the light-sensitive receptor of the rod cell and one of the best characterized GPCRs. Rhodopsin consists of the protein opsin and a covalently attached inverse agonist 11-cis-retinal. Light induces isomerization of the ligand to the agonist all-trans-retinal, resulting in the active receptor species Metarhodopsin II (Meta II). This experimental system affords the

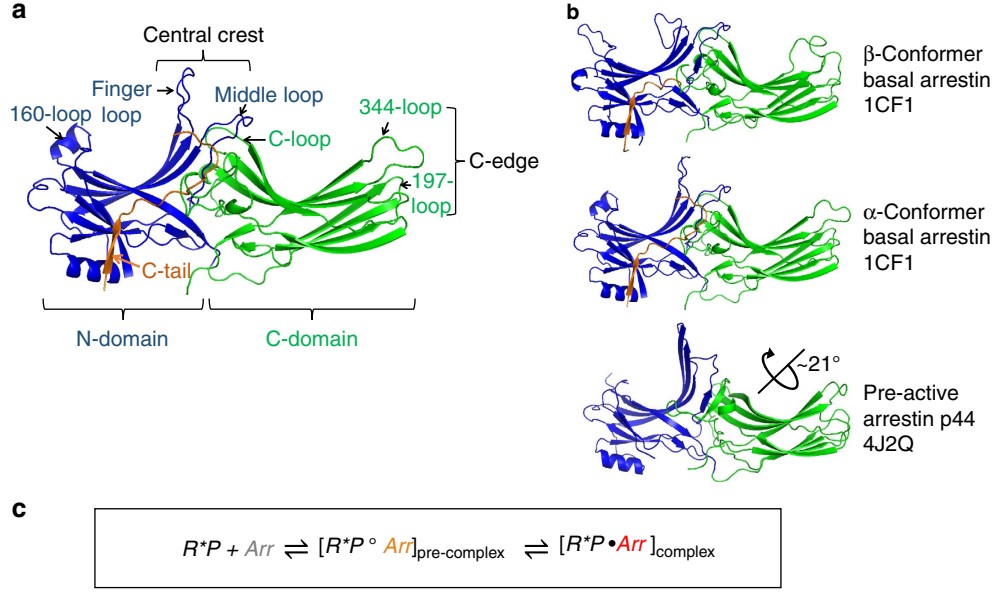

**Figure 1 | Crystal structure variants of arrestin-1 and schematic of receptor binding equilibrium.** (**a**) Structure of basal arrestin (PDB code 1CF1, molecule A)[25]. The N-domain is coloured blue, the C-domain is coloured green and the C-tail is orange. Important loops and regions referred to in this study are indicated. (**b**) Comparison of the β- and α-conformers of basal arrestin (1CF1)[25], and the C-terminally truncated pre-active arrestin p44 (4J2Q)[9]. The interdomain rotation present in p44 is indicated by the rotation axis. (**c**) Unbound basal arrestin (grey) first interacts with the phosphorylated receptor C terminus in a low-affinity pre-complex. This initial interaction primes arrestin (orange) for the conformational transition required for full activation (red) and high-affinity coupling to the active receptor.

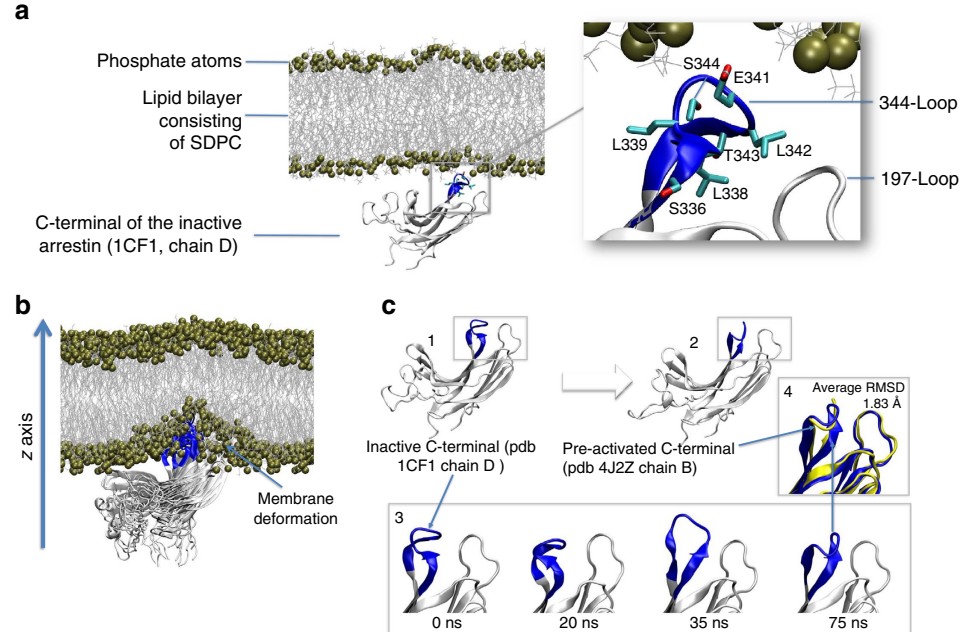

**Figure 2 | Dynamic properties of the C-edge of basal arrestin during biased and unbiased molecular dynamics simulation.** (**a**) All-atom simulation setup containing the isolated C-domain of basal arrestin (1CF1, molecule D), lipid bilayer (80 × 80 Å) composed of SDPC (1-stearoyl-2-docosahexaenoyl-sn-glycero-3-phosphocholine), water layer (not shown for simplicity) and solvated to 0.15 M NaCl, yielding a system of approximately 80,000 atoms. Inset: hydrophobic residues L338, L339 and T343 are buried between the 344-loop and the 197-loop, and polar residues E342 and S344 are directed toward the membrane. (**b**) Energetic bias of the basal conformation of the 344-loop along the z coordinate using a collective variable of the centre of mass (COM) of the C-alpha atoms of L338, L339, G340, E341 and L342 (344-loop; three replicates × 100 ns, metadynamics). The polar 344-loop is unable to penetrate the membrane and results instead in a membrane deformation. (**c**) Conformational rearrangement of the 344-loop from basal (C1) to pre-active (C2). Unbiased molecular dynamics simulation captured a structural rearrangement of the 344-loop from basal to pre-active during 100 ns (C3). Superposition (C4) of the crystal structure of pre-active arrestin p44 (yellow, 4J2Q, chain B) with the pre-active conformation obtained in simulation (blue) yields an average RMSD of 1.83 Å. The average RMSD was calculated for residues 335 to 345 and backbone atoms over the last 30 ns of the simulation MD2 (see also Supplementary Fig. 1).

advantage that pre-complex and high-affinity complex interactions can be separately observed[14,15].

In this study, molecular dynamics simulations indicate that the C-edge of active arrestin spontaneously interacts with the membrane. This observation is confirmed using fluorescence quenching methods, which further suggest distinct orientations and conformations of the C-edge membrane anchor in the pre-complex and high-affinity complex. Comparison of simulation and experimental data indicate that the C-edge orientation identified in the simulations reflects that in the high-affinity complex and is consistent with the crystal structure of the arrestin–receptor complex. In addition, we propose a C-edge conformation for the pre-complex that is congruent with fluorescence data and based on available crystal structures of arrestin. All in all, our findings present a previously unknown property of arrestin as a membrane-interacting protein.

## Results

**C-edge of pre-activated arrestin inserts into the membrane.** We first investigated how the C-domain of basal arrestin might interact with a membrane using molecular dynamics simulations (Fig. 2a). To reduce the system size, we removed both the N-domain and C-tail and simulated only the isolated C-domain. The conformational stability of the simulated isolated C-domain was steady during all simulation runs with an average RMSD (loops excluded) that did not exceed 1.7 Å compared with the crystallized C-domain of full-length arrestin (Supplementary Table 1). We observed no insertion of the arrestin C-domain into the membrane during 10 replicates with an accumulated

sampling time of 1 μs (Table 1, MD1). To explore whether loop insertion into the membrane is possible at all, we applied an energetic bias to the arrestin C-domain in the z direction toward the membrane using metadynamics (three replicates of 100 ns, Table 1, MD2). No insertion occurred, and instead a deformation of the membrane was observed (Fig. 2b). Membrane insertion of the 344-loop is not favoured in the basal state because polar residues E341 and S344 are solvent-exposed and hydrophobic residues (L338, L342 and T343) are buried between the 344-loop and the adjacent 197-loop (residues F197–P202 in bovine arrestin-1; Fig. 2a). Notably, at the end of one biased simulation from MD2 we observed a spontaneous conformational rearrangement of the 344-loop (Fig. 2c), which resembles that observed in the crystal structure of pre-activated arrestin p44 (average RMSD of 1.83 Å, Fig. 2c part 4 and Supplementary Fig. 1). In contrast to basal arrestin, hydrophobic residues of the 344-loop in p44 are solvent-exposed (Fig. 3a). We next investigated how the C-domain of p44 might interact with the membrane using unbiased molecular dynamics simulations (10 replicates of 100 ns, Table 1, MD3). In one of these simulations, we observed spontaneous insertion of the C-edge loops into the membrane (Fig. 3b). Specifically, the hydrophobic residues L338, L339 and L342 on the 344-loop as well as residues F197 and M198 on the 197-loop were embedded within the membrane interior. Apart from these hydrophobic interactions, we observed polar contacts between the C-edge loops and the membrane. In particular, S199, D200, S336, S344 and E341 established transient contacts with the membrane, which fluctuated between polar membrane head groups, solvent molecules or other polar residues in the C-domain. Notably,

**Table 1 | Details of molecular dynamics simulations.**

| Experiment ID | Simulation description | Simulation length |
|---|---|---|
| MD1 | C-domain of basal arrestin (1CF1, molecule D) Unbiased simulation | $10 \times 100\,\mu s$ |
| MD2 | C-domain of basal arrestin (1CF1, molecule D) Biased simulation well-tempered metadynamics (hills height 2, biasfactor 15) | $3 \times 100\,\mu s$ |
| MD3 | C-terminal of the pre-activated arrestin (4J2Q, chain B) Unbiased simulation | $10 \times 100\,\mu s$ |
| MD4 | C-terminal of the pre-activated arrestin (4J2Q, chain B) Biased simulation well-tempered metadynamics (hills height 2, biasfactor 15) | $3 \times 100\,\mu s$ |
| | Total simulation length | $2.6\,\mu s$ |

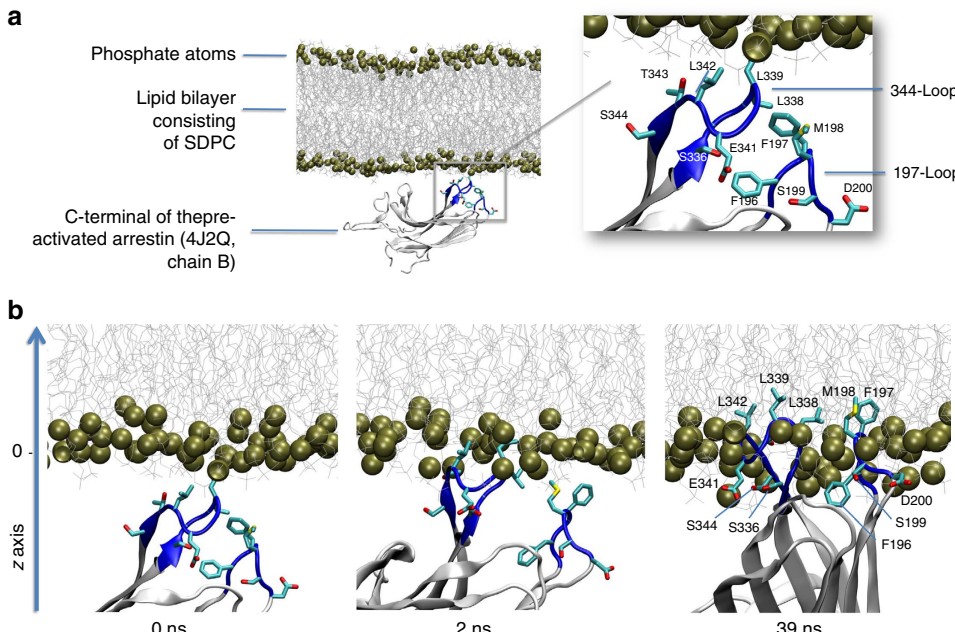

**Figure 3 | C-edge of pre-active arrestin p44 spontaneously inserts into membrane bilayer.** (**a**) All-atom simulation setup containing the isolated C-domain of the pre-active arrestin p44 (4J2Q, chain B), lipid bilayer ($80 \times 80\,Å$) composed of SDPC (1-stearoyl-2-docosahexaenoyl-sn-glycero-3-phosphocholine), water layer (not shown for simplicity) and solvated to 0.15 M NaCl yielding a system of approximately 80,000 atoms. Inset: conformation of the 344-loop in pre-active arrestin p44 directs hydrophobic residues L338, L339 and L342 toward the lipid bilayer. (**b**) In an accumulated time of $1\,\mu s$ ($10 \times 100\,ns$) of unbiased molecular dynamics simulation (MD 3), we observed one spontaneous penetration of the 344-loop into the lipid bilayer. After 40 ns simulation time, hydrophobic residues of the 344-loop (L338, L339, L342) as well as the 197-loop (F197 and M198) dip into the hydrophobic region of the lipid bilayer.

negatively charged residues were able to interact at times with negatively charged lipid head groups via a positively charged sodium ion. This was observed for D200 (197-loop) during the process of loop insertion into the membrane before undergoing a conformational change which allowed for frequent intramolecular interaction with K201 (Supplementary Fig. 2). Finally, loop insertion into the membrane was confirmed by biased simulations (three replicates of 100 ns, Table 1, MD4), which revealed an energetic well for 344-loop at $2.7 \pm 1.8\,Å$ from the polar phosphate atom layer to the centre of mass of the 344-loop ($COM_{344\text{-loop}}$) (Supplementary Fig. 3).

**Direct observation of C-edge interactions with the membrane.** We next experimentally probed the proximity of different sites on arrestin to the membrane when arrestin is in complex with phosphorylated receptor. Our approach was based on quenching of site-specifically placed fluorophores on arrestin by spin labels located at different levels within the membrane (see Supplementary Note 1). First, 16 individual sites within arrestin-1 were mutated to cysteine, and the fluorophore bimane was attached. Next, native rod outer segment membranes (ROS) containing phosphorylated rhodopsin (Rho-P) were enriched with fatty acids labelled with nitroxide spin labels (Supplementary Table 2). For N-tempoyl-palmitamide, the spin label is localized within the polar head-group region of the membrane, and for 5-doxyl-stearic acid, the spin label is localized within the hydrophobic interior of the membrane near carbon 5 on the fatty acid chain of the phospholipid. As a control, ROS membranes were also enriched with analogous fatty acids lacking spin label, specifically methyl-palmitate or stearic acid. Robust binding of all labelled arrestin mutants to the enriched ROS membranes was verified by centrifugal pull-down analysis (Supplementary Fig. 4). Our experimental conditions ensured comparable binding of arrestin mutants to both dark-state Rho-P and light-activated Rho*-P (see Supplementary Note 2).

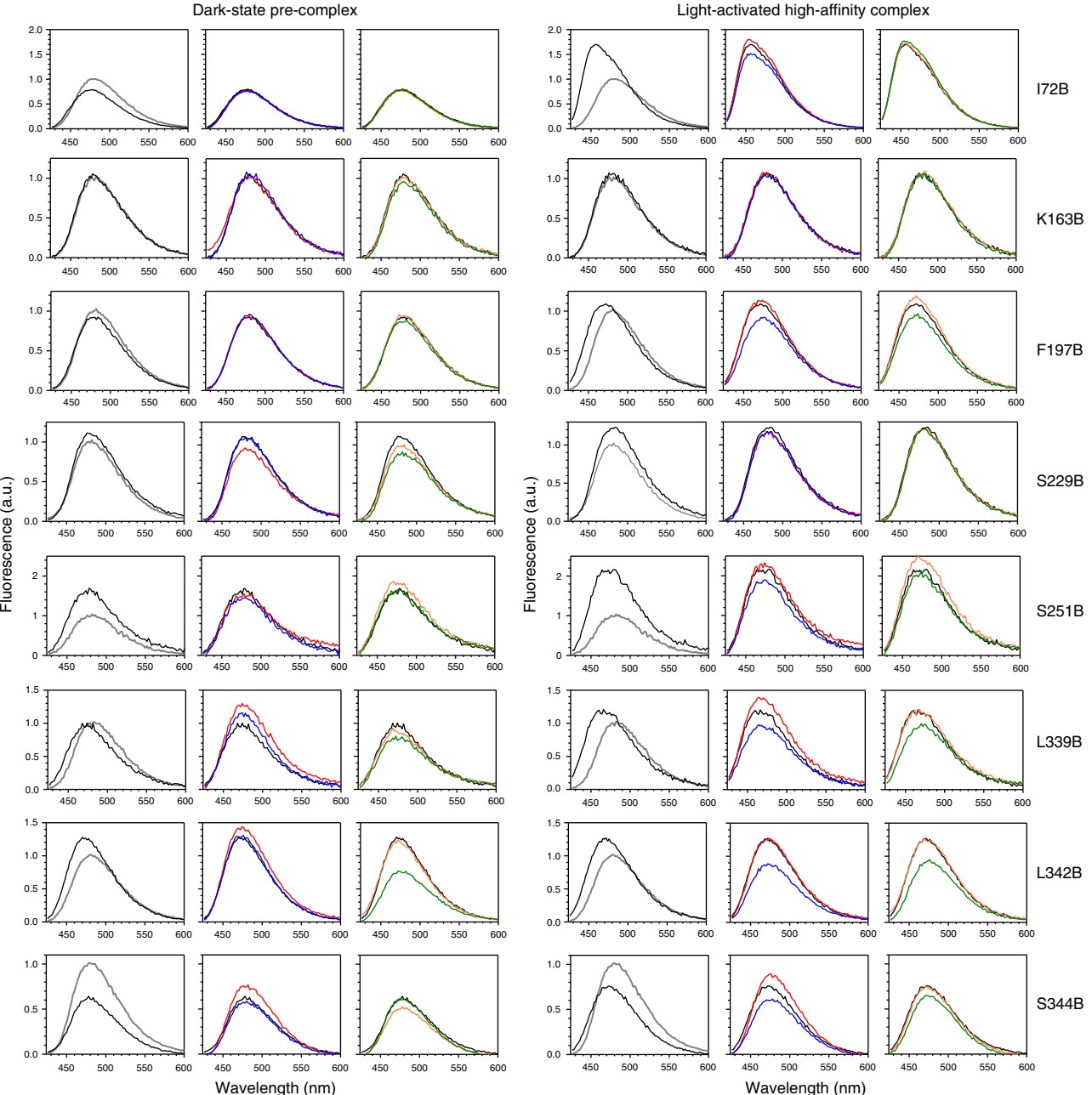

**Figure 4 | Example fluorescence spectra of bimane-labelled arrestin mutants in the presence of enriched ROS-P membranes.** The fluorescence ($\lambda_{ex}$: 400 nm) of each bimane-labelled arrestin mutant (1 μM) was measured in the unbound state (grey spectra) or when bound to ROS-P (4 μM) control membranes (black spectra). The fluorescence in the presence of fatty-acid enriched ROS-P membranes is also shown: red, methyl palmitate; blue, N-tempoyl-palmitamide; orange, stearic acid; green, 5-doxyl-stearic acid. Note that fluorophores attached to sites on the membrane anchor (197, 339, 342, 344) display a spectral blue-shift upon complex formation with ROS-P (both dark-state and light-activated), indicating localization in a hydrophobic environment. Fluorescence spectra are normalized such that the fluorescence intensity of each mutant in the unbound state equals 1.

The fluorescence of each arrestin mutant was measured in the presence of the different enriched ROS membranes (Fig. 4). Quenching efficiency was evaluated by comparing the fluorescence in the presence or absence of spin labels for both the dark-state pre-complex and the light-activated high-affinity complex (Fig. 5 and Table 2). Note that this procedure allows the exclusion of possible quenching of bimane fluorophores on arrestin by receptor tryptophan and tyrosine residues (see Supplementary Note 3). Quenching was primarily localized to the 344-loop and the 197-loop within the C-edge, as well as the 160-loop (site 161) and loops within the central crest region,

namely the middle loop (site 139) and the C-loop (site 251; see Fig. 1a). No quenching of arrestin labelled on the 344-loop was observed using a large excess of nonphosphorylated ROS membranes enriched with spin-labelled fatty acids (Supplementary Fig. 5), indicating that membrane engagement by arrestin is dependent on the presence of phosphorylated receptor.

In the pre-complex, relative deep membrane insertion was implied by significant quenching at sites 342 (30%) and 339 (15%) by 5-doxyl-stearate (Fig. 5 and Table 2). Site 342 was also quenched to a lesser extent by N-tempoyl-palmitamide

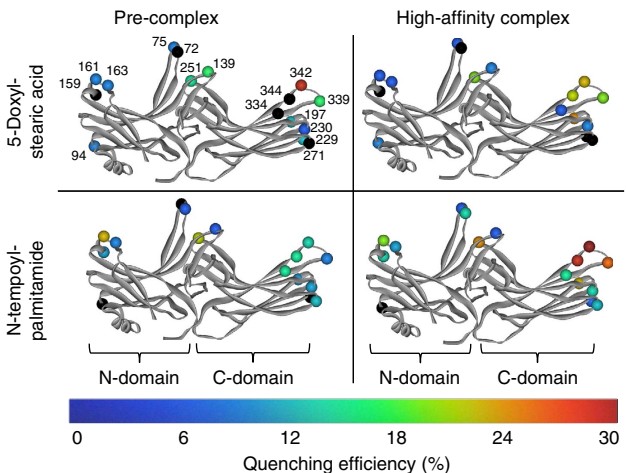

**Figure 5 | Summary of fluorescence quenching experiments.** The measured quenching efficiencies at each site on arrestin in the dark-state pre-complex and the light-activated high-affinity complex (see Table 2) is indicated by a colour spectrum ranging from 0 to 30% (spheres located at Cα). Black indicates instances where fluorescence was enhanced by the presence of spin-labelled fatty acid (that is, negative quenching). Arrestin models are based on the structure of arrestin-1 reported by Hirsch *et al.*[25] (1CF1, molecule A), and the C-tail is omitted for clarity.

(11%), along with sites 334 and 344 (13 and 17%, respectively), indicating their proximity to the polar phospholipid head-group region of the membrane. On transition to the high-affinity complex, changes in the pattern of quenching implied a shift of the C-edge closer to the head-group region. Site 342 became more accessible to *N*-tempoyl-palmitamide (31% quenching) as compared with the 5-doxyl-stearic acid (22%). Likewise, neighbouring sites 339 and 344 were more quenched by *N*-tempoyl-palmitamide (27% and 30%, respectively) than 5-doxyl-stearic acid (18% and 19%, respectively). We additionally observed strong quenching at site 197 on the 197-loop by *N*-tempoyl-palmitamide (23%) and 5-doxyl-stearic acid (23%), but only in the light-induced high-affinity complex. This result confirms the role of the 197-loop in membrane engagement as suggested by the MD simulations (Fig. 3b) and further indicates that, in the high-affinity complex, the C-domain of arrestin adopts a similar orientation as seen in the simulations.

Notably, arrestin mutants F197B, L339B, L342B and S344B displayed significant blue-shifts in fluorescence (≥10 nm) when arrestin bound to the ROS membranes (Fig. 4 and Table 2). These wavelength shifts indicate translocation of the bimane fluorophore from a polar to hydrophobic environment and are consistent with the observed quenching by 5-doxyl stearic acid at these sites. Importantly, wavelength shifts are independent of changes in fluorescence intensity and therefore confirm the membrane insertion of these sites on the C-edge. A previous EPR study of arrestin-1 indicated that spin label at site 344 is modestly immobilized in both the dark-state pre-complex and light-activated high-affinity complex[16], which is consistent with this site embedding in the highly fluid ROS phospholipid membrane[17,18]. In contrast, spin label at site 72 on the finger loop showed a considerable loss of mobility in the dark-state pre-complex and was nearly immobilized in complex with light-activated Rho*-P (ref. 16). This result is due to the embedding of site 72 within the cytoplasmic crevice of the active receptor[11,19] and is complementary to the significant blue-shift in fluorescence displayed by arrestin I72B upon binding Rho*-P (refs 11,19,20; see also Fig. 4 and Table 2).

Curiously, 'negative quenching' by 5-doxyl-stearic acid was observed for arrestin mutants N271B, T334B and S344B in the pre-complex, arising from the fact that the fluorescence in the presence of unlabelled stearic acid was lower than in the presence of the nitroxide-containing 5-doxyl-stearic acid. Carboxyl groups are known to quench fluorescence[21], and we hypothesize that this negative quenching effect is due to the carboxyl groups of the labelled and unlabelled stearic acid not being equivalent in their ability to quench bimane fluorescence. The influence of nearby carboxyl groups on bimane probes at sites 271, 334 and 344 is supported by the observed accessibility of these sites to *N*-tempoyl-palmitamide (8%, 13% and 17% quenching, respectively).

**Close proximity of other loops to the membrane surface.** Quenching by *N*-tempoyl-palmitamide was observed at site 72 (10%) on the arrestin finger loop in the high-affinity complex. In the crystal structure of the Ops*–arrestin-1 fusion complex[11] and the earlier crystal structure of Ops* bound to a peptide analogue of the finger loop[19], site 72 is located near the base of the cytoplasmic crevice of the receptor. This placement presumably makes the bimane fluorophore accessible to the spin label of *N*-tempoyl-palmitamide within the head-group region of the membrane. In contrast, no quenching by *N*-tempoyl-palmitamide was observed at site 75 in the high-affinity complex, which is consistent with the deeper placement of site 75 within the receptor crevice[11,19].

Quenching by *N*-tempoyl-palmitamide was also observed at site 251 on the arrestin C-loop in both the pre-complex (20%) and the high-affinity complex (23%). The proximity of site 251 to the membrane surface in the high-affinity complex is suggested by the crystal structure of the Ops*–arrestin-1 fusion complex, where the C-loop of arrestin is seen to be close to the cytoplasmic end of transmembrane helix 3 (ref. 11). Curiously, there was quenching of the bimane at site 251 by 5-doxyl-stearic acid in both the pre-complex (12%) and the high-affinity complex (18%), even though arrestin S251B displayed only a minimal blue-shift in fluorescence when receptor-bound (3 to 5 nm, Fig. 4 and Table 2). Likewise, the bimane at site 139 was quenched by 5-doxyl-stearic acid in the pre-complex (14%) even though no blue-shift was observed (Table 2). In the crystal structure of the Ops*–arrestin-1 complex, the alpha carbons of both sites 251 and 139 are >18 Å away from the putative position of C5 of the phospholipid acyl chain (see Fig. 6 below and the 'Methods' section for details on how membrane position was calculated for the crystal structure). Hence it is not immediately obvious how these sites could be accessible to 5-doxyl-stearate. The observed quenching of sites on the C- and middle loops by 5-doxyl-stearate might mean that these loops adopt significantly different poses than seen in the crystal structure. Alternatively, the quenching might be artefactual and due to secondary changes at these sites arising from the negatively charged head-group of the 5-doxyl-stearate.

In the crystal structure of the Ops*–arrestin-1 fusion complex[11], the 160-loop bends back over the arrestin N-domain, which allows it to interact with the cytoplasmic end of transmembrane helix 6. Although the crystal structure suggests >15 Å between the putative membrane surface and the 160-loop, we measured significant quenching at site 161 by *N*-tempoyl-palmitamide in both the pre-complex and high-affinity complex (21% and 19%, respectively). Consistently, 'negative quenching' was measured at nearby site 159 by 5-doxyl-stearate, which indicates close proximity of this site and the membrane surface (see discussion above). These results suggest the flexible 160-loop might be able to adopt other

**Table 2 | Fluorescence properties of bimane-labelled arrestin mutants in the presence of enriched rod outer segment membranes containing phosphorylated rhodopsin.**

| | Mutant | Ethanol (control) | Methyl palmitate | N-tempoyl-palmitamide | % Quenched | Stearic acid | 5-doxyl-stearic acid | % Quenched | Wavelengh shift (nm) |
|---|---|---|---|---|---|---|---|---|---|
| *Dark-state pre-complex* | | | | | | | | | |
| **N-domain** | | | | | | | | | |
| Finger loop | I72B | 0.94 ± 0.16 (2) | 0.87 ± 0.12 (2) | 0.85 ± 0.10 (2) | 2.3 | 0.90 ± 0.15 (2) | 0.94 ± 0.16 (2) | − 4.4 | − 5 |
| | M75B | 0.89 ± 0.08 (2) | 0.86 ± 0.04 (2) | 0.95 ± 0 (2) | − 10.5 | 0.86 ± 0 (2) | 0.82 ± 0.01 (2) | 4.6 | |
| Near α-helix | V94B | 1.05 ± 0 (2) | 1.02 ± 0 (2) | 1.08 ± 0.03 (2) | − 5.9 | 1.13 ± 0.03 (2) | 1.09 ± 0.04 (2) | 3.5 | |
| Middle loop | V139B | 0.86 ± 0.02 (2) | 0.87 ± 0.05 (2) | 0.87 ± 0 (2) | 0 | 0.84 ± 0 (2) | 0.72 ± 0.02 (2) | 14.3 | |
| 160-loop | V159B | 1.08 ± 0.05 (3) | 1.15 ± 0.03 (3) | 1.09 ± 0.10 (3) | 5.2 | 1.04 ± 0.10 (3) | 1.19 ± 0.02 (3) | − 14.4 | |
| | E161B | 1.3 ± 0.05 (2) | 1.4 ± 0.2 (2) | 1.1 ± 0.13 (2) | 21.4 | 1.3 ± 0.25 (2) | 1.25 ± 0.12 (2) | 3.8 | |
| | K163B | 0.92 ± 0.10 (2) | 0.98 ± 0.06 (2) | 0.94 ± 0.11 (2) | 4.1 | 0.90 ± 0.09 (2) | 0.87 ± 0.05 (2) | 3.3 | |
| **C-domain** | | | | | | | | | |
| 197-loop | F197B | 0.89 ± 0.03 (2) | 0.94 ± 0 (2) | 0.88 ± 0.06 (2) | 6.4 | 0.88 ± 0.05 (2) | 0.81 ± 0.07 (2) | 7.9 | |
| 230-loop | S229B | 1.16 ± 0.04 (2) | 1.02 ± 0.05 (2) | 1.10 ± 0 (2) | − 7.8 | 1.03 ± 0.01 (2) | 0.95 ± 0.03 (2) | 7.8 | |
| | T230B | 1.08 ± 0.17 (3) | 1.08 ± 0.14 (3) | 1.01 ± 0.15 (3) | 6.5 | 0.96 ± 0.17 (3) | 0.95 ± 0.16 (3) | 1.0 | |
| C-loop | S251B | 2.15 ± 0.5 (3) | 1.83 ± 0.31 (3) | 1.46 ± 0.10 (3) | 20.2 | 1.79 ± 0.22 (3) | 1.58 ± 0.09 (3) | 11.7 | − 3 |
| 270-loop | N271B | 1.04 ± 0.03 (3) | 1.06 ± 0.04 (3) | 0.98 ± 0.05 (3) | 7.5 | 0.89 ± 0.11 (3) | 1.05 ± 0.14 (3) | − 18.0 | |
| 344-loop | T334B | 0.70 ± 0.02 (3) | 0.64 ± 0 (3) | 0.56 ± 0 (3) | 12.5 | 0.58 ± 0.08 (3) | 0.66 ± 0.05 (3) | − 13.8 | |
| | L339B | 0.99 ± 0.02 (2) | 1.22 ± 0.09 (2) | 1.17 ± 0.07 (2) | 4.1 | 0.93 ± 0 (2) | 0.79 ± 0.04 (2) | 15.0 | − 5 |
| | L342B | 1.01 ± 0.2 (2) | 1.13 ± 0.26 (2) | 1.01 ± 0.23 (2) | 10.6 | 0.98 ± 0.21 (2) | 0.69 ± 0.08 (2) | 29.6 | − 8 |
| | S344B | 0.5 ± 0.07 (2) | 0.58 ± 0.12 (2) | 0.48 ± 0.05 (2) | 17.2 | 0.46 ± 0.02 (2) | 0.51 ± 0.04 (2) | − 10.9 | − 3 |
| *Light-activated high-affinity complex* | | | | | | | | | |
| **N-domain** | | | | | | | | | |
| Finger loop | I72B | 1.66 ± 0.13 (2) | 1.69 ± 0.07 (2) | 1.52 ± 0.12 (2) | 10.1 | 1.77 ± 0.21 (2) | 1.86 ± 0.24 (2) | − 5.1 | − 27 |
| | M75B | 0.93 ± 0.07 (2) | 0.95 ± 0.02 (2) | 0.97 ± 0.03 (2) | − 2.1 | 0.97 ± 0.02 (2) | 0.95 ± 0.05 (2) | 2.1 | − 6 |
| Near α-helix | V94B | 1.08 ± 0 (2) | 1.05 ± 0 (2) | 1.12 ± 0.04 (2) | − 6.7 | 1.23 ± 0.02 (2) | 1.18 ± 0.04 (2) | 4.1 | |
| Middle loop | V139B | 1.29 ± 0.02 (2) | 1.15 ± 0.02 (2) | 1.12 ± 0.01 (2) | 2.6 | 1.23 ± 0 (2) | 1.18 ± 0.02 (2) | 4.1 | |
| 160-loop | V159B | 1.19 ± 0.06 (3) | 1.30 ± 0.04 (3) | 1.18 ± 0.11 (3) | 9.2 | 1.19 ± 0.11 (3) | 1.38 ± 0.05 (3) | − 16.0 | |
| | E161B | 1.3 ± 0.08 (2) | 1.42 ± 0.15 (2) | 1.15 ± 0.14 (2) | 19.0 | 1.31 ± 0.26 (2) | 1.31 ± 0.12 (2) | 0 | |
| | K163B | 0.95 ± 0.10 (2) | 1.01 ± 0.05 (2) | 0.94 ± 0.11 (2) | 6.9 | 0.97 ± 0.09 (2) | 0.96 ± 0.05 (2) | 1.0 | |
| **C-domain** | | | | | | | | | |
| 197-loop | F197B | 1.07 ± 0.04 (2) | 1.13 ± 0.02 (2) | 0.87 ± 0.08 (2) | 23.0 | 1.11 ± 0.07 (2) | 0.85 ± 0.11 (2) | 23.4 | − 11 |
| 230-loop | S229B | 1.28 ± 0.03 (2) | 1.20 ± 0.03 (2) | 1.19 ± 0.02 (2) | 0.83 | 1.18 ± 0.05 (2) | 1.25 ± 0.06 (2) | − 5.9 | |
| | T230B | 1.07 ± 0.14 (3) | 1.06 ± 0.10 (3) | 0.96 ± 0.14 (3) | 9.4 | 1.03 ± 0.14 (3) | 0.98 ± 0.15 (3) | 4.8 | − 3 |
| C-loop | S251B | 2.8 ± 0.5 (3) | 2.48 ± 0.29 (3) | 1.90 ± 0.16 (3) | 23.4 | 2.39 ± 0.2 (3) | 1.95 ± 0.07 (3) | 18.4 | − 5 |
| 270-loop | N271B | 1.01 ± 0.03 (3) | 1.03 ± 0 (3) | 0.95 ± 0.06 (3) | 7.8 | 0.98 ± 0.13 (3) | 1.13 ± 0.11 (3) | − 15.3 | |
| 344-loop | T334B | 0.95 ± 0.04 (3) | 1.05 ± 0.06 (3) | 0.92 ± 0 (3) | 12.4 | 0.95 ± 0.04 (3) | 0.93 ± 0.04 (3) | 2.1 | |
| | L339B | 1.11 ± 0.11 (2) | 1.37 ± 0.08 (2) | 1.0 ± 0.03 (2) | 27.0 | 1.16 ± 0.11 (2) | 0.95 ± 0.08 (2) | 18.1 | − 10 |
| | L342B | 1.01 ± 0.23 (2) | 1.06 ± 0.21 (2) | 0.73 ± 0.16 (2) | 31.1 | 1.03 ± 0.25 (2) | 0.80 ± 0.12 (2) | 22.3 | − 10 |
| | S344B | 0.78 ± 0.04 (2) | 0.84 ± 0.02 (2) | 0.59 ± 0 (2) | 29.8 | 0.78 ± 0.08 (2) | 0.63 ± 0.05 (2) | 19.2 | − 10 |

Fluorescence intensities are normalized to the fluorescence of 1 μM unbound arrestin (see 'Methods' section for more details). Mean ± s.e. is reported. Numbers in parentheses indicate number of independent measurements. Quenching efficiencies (bold) are reported as a percentage, and the colour scheme follows the template shown in Fig. 5 (quenching levels: blue = little to none, green = moderate, red = significant). Negative wavelength shifts (bold) indicate shifts to lower values.

conformations than seen in the crystal structure of the Ops*–arrestin-1 fusion complex, specifically those that allow sites 159 and 161 close contact with the membrane surface. Interestingly, negative-stain electron microscopy of GPCR–β-arrestin complexes, in which the helical core of the receptor is not engaged by the arrestin finger loop, reveals orientations in which the tip of the N-domain is positioned close to the receptor[22,23]. Such an orientation would put the 160-loop within close proximity to the quenching group of N-tempoyl-palmitamide.

**Comparison of MD and experimental data**. To compare molecular dynamics data with quenching results in more detail, we calculated distance values from the simulations that are expected to correspond to quenching efficiencies of N-tempoyl-palmitamide and 5-doxyl-stearic acid. $Dist_P$ relates to the distances from the alpha carbons (Cα) of selected sites on arrestin to the level of the phosphate atoms (P) within the membrane. $Dist_{C5}$ corresponds to the distance of from Cα of selected sites on arrestin to the level of the carbon at position five on the second aliphatic chain (C5) of the phospholipids. We compared the $Dist_P$ and $Dist_{C5}$ averaged over the simulation (MD3) to the fluorescence quenching efficiencies observed for the high-affinity complex (Supplementary Fig. 6). MD simulations suggest that

residues 197, 339 and 342 are close ($< 5$ Å) to the phosphate atom layer, which is in line with N-tempoyl-palmitamide quenching data. One exception is site 344, for which the averaged simulation data predicted a position further away from the phosphate layer ($> 8$ Å) than that determined by fluorescence experiments. A similar discrepancy is seen for residue 344 when comparing the MD-derived $Dist_{C5}$ to the quenching efficiency of 5-doxyl-stearic acid. Differences between simulation and quenching data of the C-edge can be assigned to the fact that the isolated C-domain was simulated in the absence of the receptor, which allowed for more fluctuation. In contrast, quenching experiments were carried out in the presence of the receptor, which couples to arrestin and thereby restricts movement of the whole arrestin including the C-edge. Furthermore, the MD-derived distances were measured from Cα, while quenching data derived from a bimane probe attached to a cysteine residue, which affords a large degree of rotational freedom many angstrom units away from Cα. Despite these differences, comparison indicates that the MD simulation mirrors the position of the C-edge in arrestin-1 bound to phosphorylated Meta II (that is, high-affinity complex).

Ultimately, we selected one frame out of the pool of simulated C-domain positions and conformations based on the best fit to both the C-domain of arrestin and the membrane plane (OPM-based) in the crystallized Ops*–arrestin-1 complex (see the 'Methods' section for more details). The selected frame

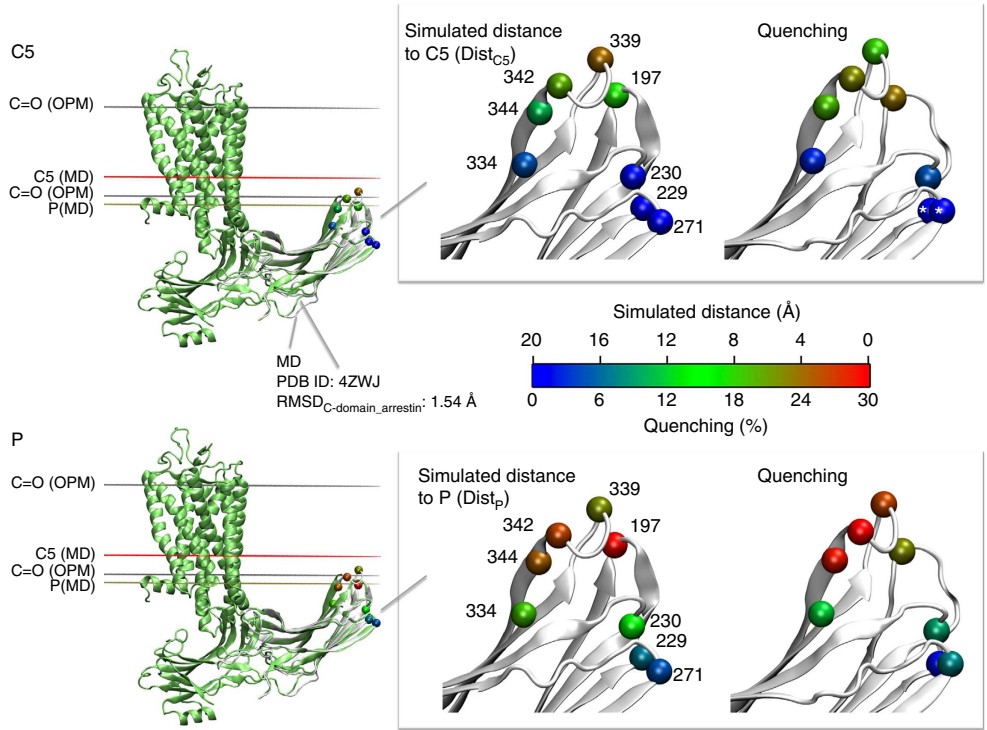

**Figure 6 | Comparison of simulation and fluorescence data with the crystal structure of the Ops\*–arrestin-1 complex.** (Left) Superimposition of the Ops\*–arrestin-1 fusion complex crystal structure (green, PDB ID: 4ZWJ) to a sampled C-edge conformation during MD simulation (grey). Coloured horizontal lines indicate the positions for individual lipid atoms: grey lines: C=O based on the OPM database, red line: carbon 5 (C5) based on MD, brown line: phosphate atoms based on MD. (Right) Distance and fluorescence maps, derived from all-atom simulation and site-directed fluorescence experiments, respectively. Distances are depicted for a sampled C-edge conformation which was selected based on the best fit to the arrestin in the Ops\*–arrestin-1 fusion complex structure with a $RMSD_{C-domain\_arrestin}$ of 1.54 Å (calculated over all backbone atoms). The colour spectrum (0 to 20 Å) corresponds to the distance of the C-alpha carbon (VdW representation) of residues on arrestin to phosphate atoms (relates to the position of the spin label in $N$-tempoyl-palmitamide) and C5 atoms (relates to the position of the spin label in 5-doxyl stearic acid). The quenching data are coloured according to the degree of quenching (0 to 30%). The white asterisk indicates instances of negative quenching (see text for details).

reflects the positioning and conformation of the crystallized C-domain with the advantage of having the C-edge loops resolved. Figure 6 presents a superposition of the sampled C-edge conformation derived from our MD simulation (MD3) to the crystallized Ops\*–arrestin-1 complex. The selected simulation frame shows that residues 197, 339, 342 and 344 are within 5 to 10 Å of the C5. Consistently, these sites were quenched by 5-doxyl-stearic acid (18 to 23%). The simulated structure also indicates that residues 229, 230, 271 and 334 are >15 Å away, which fits well to the lack of 5-doxyl-stearate-induced quenching at these sites. Furthermore, simulation data suggest that residues 197, 339, 342 and 344 are in close vicinity (<4 Å) to the phosphate (P), which is mirrored by high fluorescence quenching (>23%) at these sites by $N$-tempoyl-palmitamide. Sites 334 and 230 showed moderate quenching (~10%) with MD-estimated distances around 10 Å. Both MD and fluorescence quenching indicated a large distance between the phosphate layer and sites 229 and 271. In summary, this comparative analysis indicates that the crystal structure of the Ops\*–arrestin-1 fusion complex most likely represents the high-affinity complex.

**Membrane anchor structure in the pre-complex**. Comparison of simulation and fluorescence data suggest that the C-edge orientation and conformation obtained by MD using the p44 structure is similar to the high-affinity complex (see above). Fluorescence experiments indicate that, in the pre-complex, sites 342 and 339 are more deeply inserted than in the high-affinity complex, and site 197 is not engaged in the pre-complex

(see Fig. 5 and Table 2). Thus we anticipate that, in the pre-complex, the conformation of the C-edge and its orientation with respect to the membrane differ from that seen in the MD simulation. The 344-loop is highly flexible and can adopt multiple conformations, as seen in the different crystal structures of basal and pre-active arrestin-1 (see Fig. 1b)[9,24–27]. For example, in the crystallographic 'α-conformer' seen in the arrestin-1 crystal structure reported by Hirsch *et al.* (molecules A and C in PDB structure 1CF1), the 344-loop is extended such that the aliphatic residues L338, L339 and L342 are exposed[25]. This orientation stands in contrast to that seen in the 'β-conformer' (molecules B and D in PDB structure 1CF1), where the side chains of these residues are buried between the β-sheets of the C-edge (see Fig. 2a). Intriguingly, positioning the C-domain of the α-conformer at a similar angle to the membrane as in the high-affinity complex would allow side chains of residues 342 and 339 access to the hydrophobic interior of the membrane while excluding site 197 (Fig. 7). Such arrangement would be consistent with quenching results for the pre-complex.

## Discussion

This study describes a previously unknown function of the C-domain of arrestin-1. Membrane anchoring is an ability arrestin shares with the other major interaction partners of GPCRs, although arrestin achieves this without the lipid anchors present in G proteins and GRKs. Here we identified the 344-loop and the 197-loop as comprising the C-edge membrane anchor. Our MD and fluorescence quenching data are highly consistent

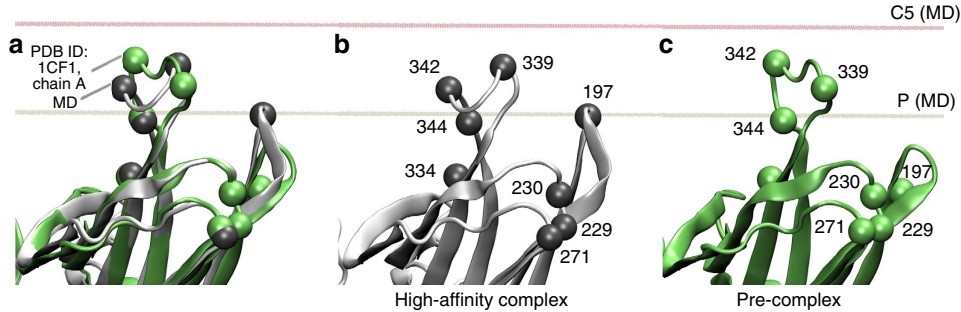

**Figure 7 | Different conformational states of the C-edge.** (**a**) Superposition of sampled MD frame obtained based on the best fitting to the crystallized Ops*arrestin-1 complex (PDB ID: 4ZWJ) to the 'α-conformer' seen in the crystal structure of arrestin-1 reported by Hirsch *et al.* (PDB ID: 1CF1, molecule A). (**b**) Sampled MD frame alone. (**c**) Crystallized 'α-conformer' alone. The averaged layer of phosphate atoms (P) and carbon 5 (C5) of lipids are simulation-based coordinates and are indicated by tan and red lines, respectively.

with one another, and together they corroborate the membrane engagement of the arrestin C-edge suggested by the crystal structure of arrestin-1 bound to active opsin[11]. Our fluorescence data further suggest a distinct membrane anchor conformation and orientation for the pre-complex, for which very little structural information exists. The membrane anchor is a key element involved in stabilizing the flexible arrestin-receptor complex along the path to tight binding. On one hand, the membrane anchor is a general binding element that bestows versatility on arrestin binding, since every GPCR resides in a membrane. On the other hand, the membrane anchor supports specificity in binding, since the C-edge is activated for membrane engagement when arrestin interacts with the phosphorylated receptor C terminus.

The C-edge membrane anchor of arrestin is markedly similar to the C2 family of membrane binding domains. These domains consist of a concave eight-stranded antiparallel β-sandwich, which interacts with the membrane using the interconnecting loops on one edge of the sandwich[28,29]. Importantly, the C-edge membrane anchor of arrestin-1 we describe here shares many of these attributes, including an abundance of basic charges within the concave surface of the β-sandwich (e.g., K235, K236, K238, K267, K332, K333), which might provide an electrostatic attractive force to bring the C-domain to the membrane. Previous studies by us and others have indicated that negatively charged phospholipids are necessary for arrestin-1 to bind light-activated phosphorylated rhodopsin[30,31]. Remarkably, acidic phospholipids are flipped to the outer membrane leaflet by rhodopsin activation[32,33]. It is interesting to consider how increased electronegativity of the membrane surface induced by rhodopsin activation might recruit the positively-charged arrestin C-edge to the membrane.

Membrane anchoring is an attribute arrestin shares with other members of the arrestin clan, whose members all share the basic arrestin fold of two near-symmetric concave beta-sandwich domains[34]. Members include the Vps26 proteins, which are part of the retromer complex involved in endosomal membrane trafficking, and the α-arrestins (also called ARRDC), which are the oldest members of the clan and are believed to function as scaffolding proteins to mediate ubiquitination and endocytic sorting of GPCRs and other receptors at the membrane[35]. All members of the arrestin clan have the capacity to interact with the membrane, although they share very little sequence identity (11–15%; ref. 35) and appear to use different mechanisms of membrane interaction[36,37].

Intriguingly, the C-edge is preserved to different extents in the four arrestins. Although the 197-loop is preserved in all members, the 344-loop is lacking in arrestin-3 (β-arrestin 2) and the short splice variant of arrestin-2 (β-arrestin 1). The longer splice variant of arrestin-2 contains an eight-amino acid insert (334LLGDLASS341) in the 344-loop[38], which is nearly identical to that in arrestin-1. Remarkably, this insert has been observed by X-ray protein crystallography to bind clathrin, and this interaction is sufficient to mediate recruitment of active β2 adrenergic receptors to clathrin-coated pits in the absence of the principal clathrin binding motif located in the C-tail[39]. Mutagenesis analysis defined the secondary clathrin-binding motif as $(L/I)_2GXL$, which contains the three membrane-interacting leucine residues (L338, L339 and L342 in arrestin-1) we have identified in this study. Interestingly, in the crystal structure of arrestin-2 bound to clathrin[39], the 344-loop bears a similar outward orientation of leucine residues as seen in the pre-activated arrestin-1 (p44)[9] (Supplementary Fig. 7). It is likely that the long variant of arrestin-2 can interact with the membrane similarly as we have observed with arrestin-1. Supposedly membrane engagement and clathrin binding are mutually exclusive, although one can imagine that sequential engagement of the membrane and clathrin by the 344-loop is conducive to endocytosis.

Regarding the other arrestin variants that lack the 344-loop, it is possible that the 197-loop is sufficient to act as a membrane anchor when arrestin engages an active GPCR, or that the shorter version of the 344-loop engages the membrane surface. Interestingly, arrestin-3 has an outwardfacing arginine residue at site 332 (also present in the shorter version of arrestin-2), which supposedly could interact with phospholipid head-groups of the membrane. Clearly, the longer version of the 344-loop is not absolutely necessary for the two β-arrestins to interact with GPCRs. No significant difference in binding of the long and short variants of arrestin-2 to the β2 adrenergic and m2 muscarinic cholinergic receptors was observed *in vitro*[40], although these variants are expressed in different tissues and cell types, suggesting they might interact with different sets of GPCRs[38,39]. It is intriguing to consider that the longer version of the 344-loop serves as variable binding element. In the visual arrestins, this loop can penetrate the membrane to stabilize both the pre-complex and high-affinity complex, while in the longer version of arrestin-2 this loop can additionally couple to clathrin. The functional role of the C-edge in the β-arrestins for binding membrane and mediating receptor internalization remains to be clarified.

The fact that the C-edge of arrestin-1 is activated for membrane binding when arrestin interacts with the phosphorylated rhodopsin C-terminus suggests the membrane anchor plays a role in the formation and stabilization of the pre-complex. The primary contact between receptor and arrestin is the binding of the phosphorylated receptor C-terminus to multiple phosphosensors (that is, basic residues) within

the N-domain. The receptor C terminus is long and flexible, and an arrestin tethered solely to the receptor C terminus would be expected to be highly mobile. The anchoring of the arrestin C-edge in the membrane would restrict this mobility, thereby bringing the receptor binding elements of arrestin (for example, finger loop) into closer proximity of the receptor and facilitating transition to the high-affinity complex. Intriguingly, electron microscopy (EM) of arrestin-2 bound to a GPCR (chimera of $\beta_2$ adrenergic and V2 vasopressin receptor) captured a 'hanging' binding mode of arrestin[23]. Considering that the receptor used in these studies was solubilized in a neutral detergent and not a phospholipid membrane, this mode of interaction could represent the pre-complex without the benefit of membrane anchoring.

A subsequent EM study has recently reported the existence of 'megaplexes' composed of a GPCR bound simultaneously to G protein and β-arrestin, which give rise to sustained signalling from internalized receptors in endosomes[22]. In these complexes, the G protein is coupled to the helical core of the receptor, and arrestin is bound nearby, supposedly tethered by the phosphorylated receptor C-terminus. The orientation of arrestin in this complex suggests deep insertion of the C-edge into the theoretical membrane plane[22], which would be expected to stabilize arrestin association within the megaplex in the absence of binding the receptor core. Notably, these EM-visualized complexes of receptor-bound β-arrestin, in which the arrestin finger loop does not engage the receptor core[22,23], could represent different possible pre-complex formations.

Our experimental findings allow us to make predictions regarding the structure of arrestin in the pre-complex and the conformational transitions that occur during formation of the high-affinity complex. We predict that the 344-loop adopts an extended, α-conformer-like conformation in the pre-complex, which allows it to deeply penetrate the hydrophobic interior of the membrane. We further expect that the C-domain is not significantly rotated in the pre-complex as compared with the basal state, which excludes the 197-loop from membrane interaction. On transition to the high-affinity complex, arrestin is fully activated and the C-domain rotates 21°, thereby allowing the 197-loop to engage the membrane. We further predict that the 344-loop adopts a folded conformation similar to that seen in the crystallized p44 structure, which favours a more shallow interaction with the membrane. This mode of binding was captured in our MD simulations using an isolated C-domain of pre-active p44, and we anticipate that this interaction significantly stabilizes the high-affinity complex. Note that the different membrane anchor orientations for the pre-complex and high-affinity complex, which we predict based on our fluorescence data, is corroborated by a recent alanine scan mutagenesis study of arrestin-1 binding to inactive and active phosphorylated rhodopsin[41].

In summary, our study redefines arrestin as both a GPCR-binding and membrane-binding protein. These binding behaviours are cooperative and dependent on one another, since arrestin cannot engage the membrane in the absence of phosphorylated receptor, and arrestin cannot bind receptors without the presence of membrane (or at least phospholipids). Our findings open new avenues of research to understand the role of the membrane in arrestin activation and receptor binding, and how these membrane interactions influence the different functions of arrestin in the cell.

## Methods

**Unbiased molecular dynamics simulation.** To study the dynamic properties of the C-edge with sufficient sampling, we focused on the isolated C-domain. The C-domain (residues 182 to 360) of the basal arrestin (1CF1, molecule D)

and the pre-activated arrestin (4J2Q, chain B) were retrieved from the Protein Databank[9,25]. The ends were capped using standard group patching (NTER and CTER). The charmm-gui membrane builder[42] was used to place the C-edge of the C-domain approximately 3 Å below the membrane. The generated membrane patch consisted of pure 1-stearoyl-2-docosahexaenoyl-sn-glyerco-3-phosphocholine (SDPC) with a size of $80 \times 80$ Å$^2$. The system was solvated and ionized to 0.15 M sodium chloride yielding a total number of atoms of $\sim 78,000$.

Molecular dynamics simulations were performed using ACEMD[43] using the following protocol: In a first stage, each system was submitted to a minimization procedure for 3,000 steps. In a second stage, the system was equilibrated using the NPT ensemble with a target pressure equal to 1.01325 bar, a time-step of 2 fs and using the RATTLE algorithm for the hydrogen atoms. In this stage, the harmonic constraints applied to the heavy atoms of the protein and ligand were progressively reduced from an initial value of 10 kcal mol$^{-1}$ Å$^{-1}$ until an elastic constant force equal to 0 kcal mol$^{-1}$ and the temperature was increased to 300 K. All the simulations were conducted using the same non-bonded interaction parameters, with a cut-off of 9 Å, a smooth switching function of 7.5 Å and the non-bonded pair list set to 9 Å. For the long-range electrostatics, we used the PME methodology with a grid spacing of 1 Å. In a third stage, production phases were performed using the NVT ensemble with aforementioned parameters but a time-step of 4 fs, and a hydrogen scaling factor of 4. This time-step is possible due to the implementation of the hydrogen mass repartitioning scheme in the ACEMD code[44]. A summary of all unbiased simulations is found in Table 1.

To elect a MD frame that best fit the Ops*P–arrestin-1 crystal structure, backbone atoms of the C-domains of all simulation frames for MD3 were first aligned to the C-domain of the crystallized Ops*–arrestin-1 complex (PDB ID: 4ZWJ), which was retrieved from the OPM database[45]. The OPM structure provides spatial arrangements of membrane proteins with respect to the hydrocarbon core of the lipid bilayer, provided as dummy atoms that describe the position of the carbonyl lipid (C=O) layer. In a second step, the C-domain-aligned frame with the best fit to the OPM-defined C=O layer was selected (see Fig. 6).

**Biased molecular dynamics simulation.** Metadynamics is a biased dynamics technique widely used to improve sampling for free energy calculations over a set of multidimensional reaction coordinates, which would not be sampled exhaustively with normal unbiased simulations[46]. It is implemented in the molecular dynamics code ACEMD using the PLUMED plugin interface[47]. We used metadynamics to estimate the energetic barrier that the C-domain requires to enter the membrane. For this, we positioned the C-domain (1CF1, molecule D) below the lipid bilayer ($80 \times 80$ Å) composed of SDPC (1-stearoyl-2-docosahexaenoyl-sn-glycero-3-phosphocholine). The system was solvated and ionized to 0.15 M NaCl, yielding a system of $\sim 80,000$ atoms. We used as reaction coordinate the $z$ position of the centre of mass (COM) of the Cα atoms of residues 338 to 342 (COM$_{344\text{-loop}}$). In addition, we set a restraining potential with an energy constant Kappa of 100 that starts acting when the 344-loop inserts deep into the membrane ($z$ value of COM$_{344\text{-loop}} > 14$ Å distance from phosphate atoms) or when the 344-loop reaches too far into the intracellular water side ($z$ value of COM$_{344\text{-loop}} > 20$ Å distance from phosphate atoms). The metadynamics parameters were set to a Gaussian hill height of 2 kcal mol$^{-1}$ with a spread of 0.2 Å for the $z$ coordinate. The deposition rate was one hill every 2 ps and a well-tempered bias factor of 15. A summary of all biased simulations is found in Table 1.

**Preparation of ROS membranes.** The ROS membranes were prepared from frozen bovine retina obtained from W.L. Lawson Company (USA). The bovines from which the retinas were obtained were slaughtered under guidelines set forth by the Humane Slaughter Act (US Public Law 85-765) and were approved for laboratory use by the State Office for Health and Social Affairs (LAGeSo Berlin). ROS isolation and rhodopsin phosphorylation using the native rhodopsin kinase was carried out as previously described[14]. Briefly, 100 thawed retina were vigorously shaken with 90 ml of cold 45% (weight to volume) sucrose in ROS buffer (70 mM potassium phosphate, 1 mM MgCl$_2$, 0.1 mM EDTA, pH 7 + 1 mM DTT and 0.5 mM PMSF), and the suspension was then centrifuged at 2,500$g$ for 5 min. The supernatant was filtered through gauze, diluted slowly 1:1 with ROS buffer, and centrifuged at 6,000$g$ for 7 min. The pellets were gently resuspended in 25.5% sucrose ($\rho = 1.105$ g ml$^{-1}$) and then layered onto four gradients of composed of 14 ml 32.25% sucrose ($\rho = 1.135$ g ml$^{-1}$) overlaid with 14 ml of 27.125% sucrose ($\rho = 1.115$ g ml$^{-1}$). The gradients were centrifuged in a swinging bucket rotor at 83,000$g$ (average centrifugal force) for 30 min. ROS were collected from the interface between the 27.12 and 32.25% solutions, diluted 1:1 with ROS buffer and pelleted by centrifugation (48,000$g$, 30 min). All the above-described steps were performed under dim red light. For rhodopsin phosphorylation, ROS were gently homogenized in the dark using an all-glass douncer in 100 mM potassium phosphate pH 7.4 (100 ml), and 8 mM ATP and 3 mM MgCl$_2$ was added. Sealed transparent tubes of this suspension were placed on a rocking platform under a standard desk lamp at room temperature. After 2 h, 50 mM hydroxylamine was added to convert all light-activated rhodopsin species to opsin. ROS were then washed three to five times by collecting the membranes by

centrifugation followed by resuspension in a generous volume of phosphate buffer. Washed membrane pellets were resuspended in a small volume of 50 mM HEPES pH 7, aliquoted, snap frozen in liquid nitrogen and stored at −80 °C. Phosphorylated opsin was regenerated to phosphorylated rhodopsin by the addition of a 3-fold molar excess of 11-cis-retinal, which was prepared in-house using HPLC[48]. Regeneration was terminated by the addition of 20 mM o-tert-butyl-hydroxylamine[49], and rhodopsin concentration was determined by the loss of 500 nm-absorbance ($\varepsilon = 0.0408 \, \mu M^{-1} \, cm^{-1}$) after bleaching in the presence of 100 mM hydroxylamine. Receptor phosphorylation level was evaluated by the 'Extra Meta II' assay, which measures the ability of arrestin to stabilize light-activated rhodopsin as Meta II (refs 14,50). In brief, ROS membranes containing regenerated phosphorylated rhodopsin were light-activated with a short flash (>500 nm) under conditions (pH 8, 2 °C) favouring Meta I ($\lambda_{max}$: 480 nm), the precursor photoproduct of Meta II ($\lambda_{max}$: 380 nm). Receptor binding by arrestin and stabilization of Meta II manifests as an increase in absorbance at 380 nm. Analysis indicated that >95% of receptors in our preparations were sufficiently phosphorylated to bind arrestin.

ROS membranes were enriched with commercially available fatty acids. 2-(3-Carboxypropyl)-4,4-dimethyl-2-tridecyl-3-oxazolidinyloxy (5-doxyl-stearic acid), methyl palmitate and stearic acid were purchased from Sigma-Aldrich, and 4-Palmitamido-2,2,6,6-tetramethylpiperidine-1-oxyl (N-tempoyl palmitamide) was purchased from Avanti Polar Lipids. Fatty acids were dissolved in ethanol to yield a stock concentration of 10 mM. ROS membranes containing phosphorylated rhodopsin (ROS-P) were diluted to 5 μM in ~2 ml of 50 mM HEPES buffer pH 7. Small aliquots of the fatty acid stocks were added to the ROS-P membranes incrementally, 2 μl every 2 min at 30 °C, followed by a 1 h incubation at 30 °C with gentle mixing. This protocol is based on that originally reported by Watts et al.[51] and was optimized by us to favour membrane insertion of fatty acids and avoid micelle formation. Fatty acids were added to a final concentration of 250 μM. As a control, ROS-P membranes were treated the same with a stock of pure ethanol (2.5% final ethanol concentration).

**Preparation of arrestin mutants.** Single cysteine mutations were introduced into a recombinant bovine arrestin-1 construct that lacks native cysteine and tryptophan residues (C63A, C128S, C143A, W194F), which is cloned into the pET15b vector for bacterial expression (Supplementary Note 4). Mutations were created using PCR and primers obtained from Sigma-Aldrich (Supplementary Table 3) and verified by sequencing (LGC genomics). The mutants used in this study were I72C, M75C, V94C, V139C, V159C, E161C, K163C, F197C, S229C, T230C, S251C, N271C, T334C, L339C, L342C and S344C. Plasmid DNA was expressed in Escherichia coli XL1-Blue supercompetent cells (Stratagene 200518) and isolated using miniprep kits (ThermoFisher Scientific). For protein expression, E.coli BL21 (DE3) competent cells (New England BioLabs) were transformed with plasmid DNA and plated on selective medium containing ampicillin. A single colony was used to inoculate 5 ml LB plus ampicillin (100 μg ml⁻¹). After 8 h incubation at 28 °C with shaking, 1 ml of this starter culture was used to inoculate 150 ml LB plus ampicillin. This culture was grown overnight at 28 °C with shaking and then split between four flasks each containing 2 litres LB plus ampicillin. The cells were induced with 30 μM IPTG once the optical density at 600 nm reached 0.6. The cells were allowed to grow >16 h at 28 °C before collecting by centrifugation (5,000g 15 min). The cells were resuspended in cold lysis buffer (10 mM Tris-HCl, 2 mM EDTA, 100 mM NaCl pH 7.5 + 5 mM DTT) and then lysed using a microfluidizer (Microfluidics) in the presence of DNAse. The lysate was centrifuged (27,000g, 30 min), and ammonium sulfate was added to the supernatant to a concentration of 0.32 g ml⁻¹. The precipitated protein was collected by centrifugation (27,000g, 30 min) and then resuspended in 10 mM Tris-HCl, 2 mM EDTA pH 7.0 + 5 mM DTT. After another centrifugation step to remove insoluble protein, the lysate was filtered through a 0.8 μm cellulose filter and loaded onto 3 × 5 ml HiTrap heparin columns (GE Healthcare), while diluting 1:3 with 10 mM Tris-HCl, 2 mM EDTA, 100 mM NaCl pH 7 + 5 mM DTT. After loading, the column was washed with the same buffer, and arrestin was eluted with a NaCl gradient (0.1–0.5 M). Arrestin-containing fractions were determined by SDS–PAGE, pooled, filtered (0.22 μm) and 5 mM DTT was added. This protein was then loaded onto a 5 ml HiTrap SP column coupled to a HiTrap Q column (GE Healthcare) while diluting 1:10 with 10 mM Tris-HCl, 2 mM EDTA pH 8.5 + 5 mM DTT. After loading, the SP column was removed, and the Q column was washed with pH 8.5 buffer, and arrestin was eluted with a two-step NaCl gradient, 0–0.1 M and 0.1–0.5 M. The arrestin-containing fractions were again determined by SDS PAGE, pooled, concentrated, and buffer exchanged against 50 mM HEPES, 130 mM NaCl pH 7. Protein concentration was determined by absorbance at 280 nm ($\varepsilon = 0.02076 \, \mu M^{-1} \, cm^{-1}$). Aliquots of purified protein were snap frozen in liquid nitrogen and stored at −80 °C.

For fluorescent labelling, the purified arrestin was thawed and diluted to 20 to 50 μM in 50 mM HEPES, 130 mM NaCl pH 7. An ~100 mM stock of monobromobimane (ThermoFisher Scientific) was prepared in dimethyl sulfoxide, and the concentration was determined by the absorbance at 380 nm ($\varepsilon = 0.005 \, \mu M^{-1} \, cm^{-1}$, 1:1,000 dilution of stock in ethanol). Monobromobimane was added to arrestin at 1:50 molar excess and incubated in the dark at room temperature. After 1 h, an additional 50-fold molar excess of fluorophore was added, and labelling proceeded for an additional 2 h. Free label was removed by

washing using centrifugal filter devices (Amicon Ultra-0.5), followed by size exclusion chromatography using microcolumns prepared with Sephadex G15 (Sigma). Concentration and labelling efficiency was determined by absorbance (using extinction coefficients described above). Note that bimane contributes absorbance at 280 nm roughly equal to its absorbance at 380 nm, and this value must be subtracted from the protein absorbance peak before calculating the arrestin concentration.

**Centrifugal pull-down analysis.** The ability of each bimane-labelled arrestin mutant to bind phosphorylated rhodopsin in the dark state (pre-complex) and following light activation (high-affinity complex) was evaluated using a centrifugal pull-down assay. Briefly, 1 μM bimane-labelled arrestin mutant was mixed with ROS membranes containing phosphorylated rhodopsin (4 μM; 100 μl volume, 50 mM HEPES pH 7). The samples were either kept in the dark or light-activated (>495 nm) for 15 s, followed by centrifugation at 16,000g for 10 min. The supernatant was removed, and the pellets were solubilized in loading buffer containing 2% SDS and subjected to SDS–PAGE. Bands were visualized using Coomassie Brilliant Blue. Binding to ROS membranes enriched with the different fatty acids was analysed alongside ROS membranes lacking fatty acids (Supplementary Fig. 4).

**Fluorescence spectroscopy.** Steady-state fluorescence was measured using a SPEX Fluorolog (1680) instrument in front-face mode. The samples were excited at 400 nm, and emission was collected at 420–600 nm (2 nm step size, 0.5 s integration per point). Excitation slits were narrowed to 0.3 mm to minimize light-activation of the rhodopsin, and emission slits were widened to 4 mm. Samples generally contained 1 μM labelled arrestin and fatty-acid-enriched ROS containing 4 μM phosphorylated rhodopsin (50 mM HEPES pH 7.0, 20 °C), and the spectra were measured in the dark state and following light activation. The samples were light-activated for 10 s using a 150 W fibre optic light source filtered through a heat filter (Schott KG2) and a >495 nm long-pass filter. The integrated fluorescence intensity of each spectrum was calculated using the programme Sigma Plot 13.0 after subtracting appropriate background spectra. For each mutant, fluorescence spectra in the presence of ROS were normalized to the spectrum of 1 μM unbound arrestin, such that the maximal intensity was equal to 1 (Fig. 4 and Table 2). Quenching efficiencies were calculated by comparing the fluorescence intensity in the presence of the nitroxide-labelled fatty acids to the fluorescence with the corresponding unlabelled fatty acid (see Supplementary Table 1).

**Data availability.** The authors declare that data supporting the findings of this study are available within the paper and its Supplementary Information files and from the corresponding authors (M.E.S. and J.S.) upon reasonable request. The following PDB accession codes were used in this work (1CF1, 4J2Q, 4ZWJ).

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

## Acknowledgements

This work was supported by grants from the Deutsche Forschungsgemeinschaft (SO1037/1-2 to M.E.S.), the Berlin Institute of Health (Delbrück Fellowship BIH_-PRO_314 to M.E.S.) and the Instituto de Salud Carlos III, El Fondo Europeo de Desarrollo Regional (FEDER) (CP12/03139 and PI15/00460 to J.S.). M.E.S. and J.S. participate in the European COST Action CM1207 (GLISTEN).

## Author contributions

M.E.S. and J.S. designed the experiments; J.S. carried out the molecular dynamics simulations; B.B. expressed and purified arrestin mutants; C.C.M.L. and M.E.S. prepared the ROS membranes and fluorescently labelled arrestin mutants, performed functional assays and carried out the fluorescence experiments; C.C.M.L., M.E.S. and J.S. analysed the data; M.E.S. and J.S. wrote the paper.

## Additional information

**Competing financial interests:** The authors declare no competing financial interests.

