## [Peer Review File · Nature Communications]

Reviewer #1 (Remarks to the Author)

The authors focused on the aspect of the arrestin-receptor interaction that was largely neglected: direct arrestin interactions with the membrane. To address this issue, using existing crystal structures, they performed molecular dynamics simulations, which suggested that in its "active", but not basal conformation, arrestins have a potential to directly bind the membrane via the distal edge of the C-domain. The authors should be commended for extensive follow-up experimental studies with bimane-labeled arrestin-1 mutants, which to a large extent support the results of the simulations. It is likely that the cases where the authors detected discrepancies between the predictions and experimental data, these might be explained by the quenching of bimane fluorescence by Trp residues pretty abundant in rhodopsin. The authors should look into this possibility.

Overall, the study is of very high quality and considerable interest to the scientific community, as arrestins appear to be important players in GPCR signaling. However, several experimental and presentation issues should be addressed.

Substantive issues:

1. The authors should consider possible quenching by the tryptophanes in rhodopsin. This might explain some of the data. This possibility should be mentioned and discussed.
2. One study used EPR to compare arrestin-1 binding to inactive and active phosphorylated rhodopsin (using authors' terms, these correspond to the pre-complex and high-affinity complex) (Proc Natl Acad Sci U S A. 2006 Mar 28;103(13):4900-5). The authors should compare their data and predictions with the results of that study.
3. Suppl Fig S4. The data suggest relatively small difference between arrestin-1 binding to dark (inactive) and light-activated phospho-rhodopsin, in contrast to high specificity of arrestin-1 for active phosphorhodopsin reported in assays using 1-2 nM arrestin (J Biol Chem. 1999 Apr 23;274(17):11451-4). Considering the concentrations of the two proteins used here and the affinity of arrestin-1 for inactive phospho-rhodopsin determined earlier (Proc Natl Acad Sci U S A. 2013 Jan 15;110(3):942-7), this is likely what can be expected. The authors should explicitly discuss this for the benefit of the readers less familiar with arrestins.

Presentation issues:

4. For an average reader the authors should explain the physics of bimane fluorescence quenching by a spin label (possibly in the supplement).
5. In many GPCRs GRK phosphorylation sites are not in the C-terminus, but in other parts of the receptor (e.g., in M2 muscarinic receptors, which is often used as a model, GRKs phosphorylate the 3rd cytoplasmic loop).
6. B-arrestins bind and bring to the receptor many signaling proteins in addition to kinases (e.g., ubiquitin ligases, PDE4, and many other proteins; see Proc Natl Acad Sci U S A. 2007 Jul 17;104(29):12011-6).
7. There are a few typos that need to be corrected: p. 9, "On exception" was meant to be "One exception"; etc.

Reviewer #2 (Remarks to the Author)

General Comment:

In this manuscript Lally et al characterize through molecular simulations and fluorescence spectroscopy of arrestin-1 and rhodopsin the C-edge engagement of arrestin-1 with the membrane. They identify that the 344 loop and the 197 loop comprise an anchor and that this engagement changes between the pre complex and activated complex of arrestin-1. Much of the rationale for this study as indicated by the authors is from both their previous fluorescence studies and the recent crystal structure determination of the Opsin/arrestin-1 fusion complex. Interestingly, as the authors point out, the loop is not completely resolved and no membrane is

present in the crystallography studies. Therefore, the role of the 344 loop and C-edge remained to be determined in an alternative context.

Two relevant questions not addressed in this present study are what the minimally appropriate context should be in which to characterize the loops and C-edge, and do the findings obtained in the study have functional significance for receptor signaling beyond membrane anchoring. The impact of the manuscript would be significantly strengthened if these questions were addressed.

In a study by Xiao K1, et al, ("Functional specialization of beta-arrestin interactions revealed by proteomic analysis Proc Natl Acad Sci U S A. 2007 Jul 17;104(29):12011-6"), it is determined that arrestin2 and arrestin3 interact with more than 100 proteins, and many of them are membrane associated. Moreover, arrestin1 is found outside the visual system and as demonstrated by Oakley RH et al, ("Differential affinities of visual arrestin, beta arrestin1, and beta arrestin2 for G protein-coupled receptors delineate two major classes of receptors. Biol Chem. 2000. Jun 2;275(22):17201-10 "), arrestin1 can interact similarly to arrestin2 and arrestin3 with membrane bound G protein coupled receptors other than rhodopsin. Thus, the C-loops and edges of arrestin1 could just as well interact with sites in membrane bound proteins and the determinants that the authors identified could provide protein binding motifs as well as membrane anchors. A direct and fairly simple method to investigate this would be to determine in a few instances the effect of mutagenesis on the kinetics and dynamics of arrestin1 interactions with an activated receptor such as the V2R as described in the above reference, using microscopy, BRET, FRET, and/or other complementary methods for measuring protein-protein interaction dynamics in a model cell system.

Minor comment:

In figure 1a add the rotation axis for the 20 degree change.

Reviewer #3 (Remarks to the Author)

In this work, Lally et al. use molecular dynamics simulations and site-directed fluorescence spectroscopy to study the nature of the complex between the GPCR rhodopsin and its regulatory protein arrestin-1 during receptor activation. The authors suggest that loops within the C-edge of activated arrestin function as a membrane anchor, and the precise nature of the arrestin-membrane interactions are different between the low-affinity pre-complex and the high-affinity complex.

This work nicely complements recent findings in the topic of the molecular basis of GPCR-arrestin interactions, such as the X-ray structure of the rhodopsin-arrestin complex (Kang, Y., Zhou, X. E., Gao, X., He, Y., Liu, W., Ishchenko, A., et al. (2015). Crystal structure of rhodopsin bound to arrestin by femtosecond X-ray laser. *Nature*, 523(7562), 561-567. <http://doi.org/10.1038/nature14656>) or the recent indications that GPCRs, G proteins and arrestins may form "super-complexes" (Thomsen, A. R. B., Plouffe, B., Cahill, T. J., III, Shukla, A. K., Tarrasch, J. T., Dosey, A. M., et al. (2016). GPCR-G Protein-Arrestin Super-Complex Mediates Sustained G Protein Signaling. *Cell*, 166(4), 907-919. <http://doi.org/10.1016/j.cell.2016.07.004>). In this regard, while the authors discuss the rhodopsin-arrestin structure in the light of their data, they do not discuss the "super-complex", surely because this work has been published very recently. In my opinion, the present manuscript should include a careful discussion on how their data may fit with the possible existence of "super-complexes".

While the authors have previously hypothesized the existence of such arrestin-membrane interactions during rhodopsin activation (Peterhans, C., Lally, C. C. M., Ostermaier, M. K., Sommer, M. E., & Standfuss, J. (2016). Functional map of arrestin binding to phosphorylated opsin, with and without agonist. *Scientific Reports*, 6, 28686-15. <http://doi.org/10.1038/srep28686>; Sommer, M. E., Hofmann, K. P., & Heck, M. (2012). Distinct loops in arrestin differentially regulate ligand binding within the GPCR opsin. *Nature*

Communications, 3, 995. <http://doi.org/10.1038/ncomms2000>), in this work they provide convincing additional experimental evidence, complemented by a relevant computational analysis.

In my opinion, this is a timely piece of work, well written, and dealing with a topic of great interest to researchers in the fields of GPCRs and cellular signaling. Its ideas nicely complement recent high-profile publications, and will likely influence thinking in the field. The experimental and computational methods are adequate for the work, and are clearly explained.

In summary, I recommend publication of this manuscript in Nature Communications, with minor modifications (detailed below).

a. Include a discussion on how the presented data fit with the recently published work on the "super-complex" between a GPCR, a G protein, and an arrestin.

b. In Table 2, would it be possible to group (or label) the residues according to which domain they belong (i.e. 344-loop, 197-loop, etc...). Also, if the significant changes in fluorescence were highlighted (perhaps with a simplified version of the color code used in Figure 5), this would help enormously to read the data in the table.

c. The molecular dynamics simulations were performed on the isolated C-domain of arrestin. How did this severe truncation affect the structure/dynamics of the protein? I would expect that the authors show some data extracted from the simulation trajectories showing convincingly that the isolated C-domain has a similar structure and dynamic behavior than the full arrestin.

Reviewers' comments:

Reviewer #1 (Remarks to the Author):

The authors focused on the aspect of the arrestin-receptor interaction that was largely neglected: direct arrestin interactions with the membrane. To address this issue, using existing crystal structures, they performed molecular dynamics simulations, which suggested that in its "active", but not basal conformation, arrestins have a potential to directly bind the membrane via the distal edge of the C-domain. The authors should be commended for extensive follow-up experimental studies with bimane-labeled arrestin-1 mutants, which to a large extent support the results of the simulations. It is likely that the cases where the authors detected discrepancies between the predictions and experimental data, these might be explained by the quenching of bimane fluorescence by Trp residues pretty abundant in rhodopsin. The authors should look into this possibility.

Overall, the study is of very high quality and considerable interest to the scientific community, as arrestins appear to be important players in GPCR signaling. However, several experimental and presentation issues should be addressed.

Substantive issues:

1. The authors should consider possible quenching by the tryptophanes in rhodopsin. This might explain some of the data. This possibility should be mentioned and discussed.

Although nearby tryptophans or tyrosines on rhodopsin could quench the fluorescence of bimanies attached to arrestin, we do not believe this possibility had an influence on our experiments. Firstly, the cytoplasmic face of rhodopsin is fairly lacking in Trp and Tyr residues. The closest candidates are Y60, Y74, and Y136 (bovine rhodopsin numbering), and these are located many angstroms up the helical bundle. Inspection of the Ops*/arrestin-1 fusion complex crystal structure shows that no Trp or Tyr residues on the receptor are in close proximity to sites on arrestin to which we attached bimane fluorophores. For example, Y136 on opsin is nearly 40 Å away from site 344 on the C-edge of arrestin. Secondly, our experiments were designed such that if there was a quenching effect of receptor Trp/Tyr residues on bimanies on arrestin, we would have observed this in control measurements using ROS membranes that contained no spin-labelled fatty acids (SLFA). In determining the quenching effect of SLFA, we compared the fluorescence in the presence of the SLFA to the fluorescence in the absence of SLFA. Any quenching by receptor Trp/Tyr residues would be the same between these two measurements and hence would not influence the observed quenching by SLFA.

As suggested, we have added text to page 6 of the revised manuscript, as well as Supplementary Note 3, to explain why Trp/Tyr quenching likely does not affect our fluorescence experiments.

2. One study used EPR to compare arrestin-1 binding to inactive and active phosphorylated rhodopsin (using authors' terms, these correspond to the pre-complex and high-affinity complex) (Proc Natl Acad Sci U S A. 2006 Mar 28;103(13):4900-5). The authors should compare their data and predictions with the results of that study.

In this EPR study by Hubbell and co-workers, spin label at site 344 on arrestin showed a slight decrease in mobility upon pre-complex formation with dark-state Rho-P, which did not change much upon light-activation. Site 197 showed no change in mobility in the pre-complex or high-affinity complex. The fact that spin labels on the C-edge did not show a change in mobility supports their localization in the membrane, since the rod outer segment phospholipid membrane is highly fluid and dynamic (Cone *Nature New Biol* 1972) due to a large concentration of docosahexanoic acid (Anderson and Maude *Biochem* 1970) and is not expected to have a significant effect on spin label mobility. Our current study goes along with the EPR study as the localization of the C-edge in

the hydrophobic membrane is reflected in the blue-shifts in fluorescence of the bimane fluorophores..

In contrast, spin label at site 72 on the finger loop showed a massive loss of mobility upon light activation of Rho-P in the EPR study due to it being buried within the cytoplasmic crevice of the active receptor. This fact is also reflected in our fluorescence experiments, where bimane attached to site 72 showed a large blue-shift and increase in fluorescence due to the hydrophobicity of the receptor crevice.

In summary, spin labels report on mobility and bimane reports on changes in solvent polarity. Our fluorescence results indicated that both the C-edge and the finger loop embed in hydrophobic environments, and the EPR data indicate the differences in these environments (fluid membrane *versus* rigid interior of a protein).

We thank the reviewer for this suggestion, as it strengthens the interpretation of our fluorescence results. As requested, we have added text discussing the EPR study on page 7 of the revised manuscript.

3. Suppl Fig S4. The data suggest relatively small difference between arrestin-1 binding to dark (inactive) and light-activated phospho-rhodopsin, in contrast to high specificity of arrestin-1 for active phosphorhodopsin reported in assays using 1-2 nM arrestin (J Biol Chem. 1999 Apr 23;274(17):11451-4). Considering the concentrations of the two proteins used here and the affinity of arrestin-1 for inactive phospho-rhodopsin determined earlier (Proc Natl Acad Sci U S A. 2013 Jan 15;110(3):942-7), this is likely what can be expected. The authors should explicitly discuss this for the benefit of the readers less familiar with arrestins.

Arrestin-1 binding affinity (Kd) for Rho*-P has been reported to be in the nanomolar range (Pulvermüller et al. *Biochem* 1997; Bayburt et al. *JBC* 2011). For dark-state Rho-P, Zhuang et al. (*PNAS* 2013) reported a Kd of ~80 µM, which was measured at 100 mM NaCl. In the absence of salt, our own group measured a Kd around 1 µM for arrestin association with dark-state Rho-P (Sommer, Hofmann, Heck *Nat Comms* 2012). The difference between these values is explained by differences in receptor phosphorylation level and the different salt concentrations. Binding to dark-state Rho-P is primarily electrostatic and therefore strengthened in the absence of salt. As suggested we have added Supplementary Note 2 (referred to on page 6 of the revised manuscript) to explain how comparative levels of binding to dark-state Rho-P and Rho*-P were achieved.

Presentation issues:

4. For an average reader the authors should explain the physics of bimane fluorescence quenching by a spin label (possibly in the supplement).

We have added a basic review of the quenching mechanism as Supplementary Note 1.

5. In many GPCRs GRK phosphorylation sites are not in the C-terminus, but in other parts of the receptor (e.g., in M2 muscarinic receptors, which is often used as a model, GRKs phosphorylate the 3rd cytoplasmic loop).

We have modified the text in the introduction to include this fact.

“The active receptor is also phosphorylated on multiple sites on its C-terminus or cytoplasmic loops by GPCR kinases...”

6. B-arrestins bind and bring to the receptor many signaling proteins in addition to kinases (e.g., ubiquitin ligases, PDE4, and many other proteins; see Proc Natl Acad Sci U S A. 2007 Jul 17;104(29):12011-6).

We have modified the introduction to include this fact, and cited the suggested proteomic study.

“The β-arrestins interact additionally with hundreds of other proteins with a wide array of functions, including signalling kinases and phosphatases, ubiquitin ligases, transcription factors, cytoskeletal elements, and many more⁶. The β-arrestins mediate their own signalling networks^{7,8}.”

7. There are a few typos that need to be corrected: p. 9, "On exception" was meant to be "One exception"; etc.

We have done our best to correct all typos.

Reviewer #2 (Remarks to the Author):

General Comment:

In this manuscript Lally et al characterize through molecular simulations and fluorescence spectroscopy of arrestin-1 and rhodopsin the C-edge engagement of arrestin-1 with the membrane. They identify that the 344 loop and the 197 loop comprise an anchor and that this engagement changes between the pre complex and activated complex of arrestin-1. Much of the rationale for this study as indicated by the authors is from both their previous fluorescence studies and the recent crystal structure determination of the Opsin/arrestin-1 fusion complex. Interestingly, as the authors point out, the loop is not completely resolved and no membrane is present in the crystallography studies. Therefore, the role of the 344 loop and C-edge remained to be determined in an alternative context.

Two relevant questions not addressed in this present study are what the minimally appropriate context should be in which to characterize the loops and C-edge, and do the findings obtained in the study have functional significance for receptor signaling beyond membrane anchoring. The impact of the manuscript would be significantly strengthened if these questions were addressed.

In a study by Xiao K. et al., ("Functional specialization of beta-arrestin interactions revealed by proteomic analysis" Proc Natl Acad Sci U S A. 2007 Jul 17;104(29):12011-6"), it is determined that arrestin2 and arrestin3 interact with more than 100 proteins, and many of them are membrane associated. Moreover, arrestin1 is found outside the visual system and as demonstrated by Oakley RH et al, ("Differential affinities of visual arrestin, beta arrestin1, and beta arrestin2 for G protein-coupled receptors delineate two major classes of receptors" Biol Chem. 2000. Jun 2;275(22):17201-10 "), arrestin1 can interact similarly to arrestin2 and arrestin3 with membrane bound G protein coupled receptors other than rhodopsin. Thus, the C-loops and edges of arrestin1 could just as well interact with sites in membrane bound proteins and the determinants that the authors identified could provide protein binding motifs as well as membrane anchors. A direct and fairly simple method to investigate this would be to determine in a few instances the effect of mutagenesis on the kinetics and dynamics of arrestin1 interactions with an activated receptor such as the V2R as described in the above reference, using microscopy, BRET, FRET, and/or other complementary methods for measuring protein-protein interaction dynamics in a model cell system.

Although we whole-heartedly agree with the reviewer that the membrane anchor of arrestin should be further characterized as to its potential role in receptor and/or arrestin signalling, we believe these studies are beyond the scope of the current study. Our paper describes a biophysical structure-based characterization of arrestin-membrane interactions. We present the first direct evidence that arrestin has a membrane anchor and provide significant insight into the structure and function of the membrane anchor in formation of the arrestin-receptor complex. We believe our experimental approach is necessary and sufficient for a first description of this undoubtedly important functional domain of arrestin.

The ideal choice of system for our experiments was arrestin-1 and rhodopsin, the best-described model system for studying arrestin-GPCR interactions. The greatest advantage is that large quantities of receptor, phosphorylated by the native kinase and residing in the native membrane, can be obtained for biophysical measurements. In addition, arrestin-1 has been extensively mutated for site-directed spectroscopic studies, so the effects of mutation and probe attachment can be anticipated and easily assessed. Furthermore, the landmark first crystal structure of an

arrestin-receptor complex is of arrestin-1 and rhodopsin, which allows our results to be straightforwardly interpreted in a structural context.

Although arrestin-1 has been reported to interact with some non-receptor binding partners like AP2, JNK3, MDM2, and NSF, the β -arrestins will be much better candidates for studying the role of the C-edge membrane anchor in receptor/arrestin signalling *in vivo*.

We anticipate that our study will open new avenues of research into the role of the membrane in arrestin-receptor interactions and signalling. Experiments similar to those suggested by the reviewer will undoubtedly be taken up by us and others in the near future. We are certainly aware of the potential implications of the C-edge to interact with membrane-associated proteins, or to act as a scaffold to bring proteins to the membrane. As we discuss in our paper, the 344-loop binds clathrin (Kang et al. *JBC* 2009), and we hypothesize how this interaction might facilitate receptor internalization.

In summary, we agree with the reviewer regarding the direction of this new chapter in arrestin research. In this respect, we believe that our study represents a first important milestone describing the previously unknown membrane anchor of arrestin that will inspire the research community to perform follow-up studies regarding the membrane association of arrestin and its role for signalling in more cell-based assays and for different receptor types.

Minor comment:

In figure 1a add the rotation axis for the 20 degree change.

We have added a rotation axis near the C-domain of the p44 structure shown in panel 1b to indicate the movement of the C-domain in comparison to the other arrestin structures

Reviewer #3 (Remarks to the Author):

In this work, Lally et al. use molecular dynamics simulations and site-directed fluorescence spectroscopy to study the nature of the complex between the GPCR rhodopsin and its regulatory protein arrestin-1 during receptor activation. The authors suggest that loops within the C-edge of activated arrestin function as a membrane anchor, and the precise nature of the arrestin-membrane interactions are different between the low-affinity pre-complex and the high-affinity complex.

*This work nicely complements recent findings in the topic of the molecular basis of GPCR-arrestin interactions, such as the X-ray structure of the rhodopsin-arrestin complex (Kang, Y., Zhou, X. E., Gao, X., He, Y., Liu, W., Ishchenko, A., et al. (2015). Crystal structure of rhodopsin bound to arrestin by femtosecond X-ray laser. *Nature*, 523(7562), 561-567. <http://doi.org/10.1038/nature14656>) or the recent indications that GPCRs, G proteins and arrestins may form "super-complexes" (Thomsen, A. R. B., Plouffe, B., Cahill, T. J., III, Shukla, A. K., Tarrasch, J. T., Dosey, A. M., et al. (2016). GPCR-G Protein-Arrestin Super-Complex Mediates Sustained G Protein Signaling. *Cell*, 166(4), 907-919. <http://doi.org/10.1016/j.cell.2016.07.004>). In this regard, while the authors discuss the rhodopsin-arrestin structure in the light of their data, they do not discuss the "super-complex", surely because this work has been published very recently. In my opinion, the present manuscript should include a careful discussion on how their data may fit with the possible existence of "super-complexes".*

*While the authors have previously hypothesized the existence of such arrestin-membrane interactions during rhodopsin activation (Peterhans, C., Lally, C. C. M., Ostermaier, M. K., Sommer, M. E., & Standfuss, J. (2016). Functional map of arrestin binding to phosphorylated opsin, with and without agonist. *Scientific Reports*, 6, 28686-15. <http://doi.org/10.1038/srep28686>; Sommer, M. E., Hofmann, K. P., & Heck, M. (2012). Distinct loops in arrestin differentially regulate ligand binding within the GPCR opsin. *Nature**

Communications, 3, 995. <http://doi.org/10.1038/ncomms2000>), in this work they provide convincing additional experimental evidence, complemented by a relevant computational analysis.

In my opinion, this is a timely piece of work, well written, and dealing with a topic of great interest to researchers in the fields of GPCRs and cellular signaling. Its ideas nicely complement recent high-profile publications, and will likely influence thinking in the field. The experimental and computational methods are adequate for the work, and are clearly explained.

In summary, I recommend publication of this manuscript in Nature Communications, with minor modifications (detailed below).

a. Include a discussion on how the presented data fit with the recently published work on the "super-complex" between a GPCR, a G protein, and an arrestin.

We have added a text to the Results section p. 9 and the Discussion p. 15, in which we interpret our results in light of the recent report on Megaplexes. Notably, we hypothesize on how these EM data may represent possible pre-complex formations.

b. In Table 2, would it be possible to group (or label) the residues according to which domain they belong (i.e. 344-loop, 197-loop, etc...). Also, if the significant changes in fluorescence were highlighted (perhaps with a simplified version of the color code used in Figure 5), this would help enormously to read the data in the table.

We have modified Table 2 as suggested.

c. The molecular dynamics simulations were performed on the isolated C-domain of arrestin. How did this severe truncation affect the structure/dynamics of the protein? I would expect that the authors show some data extracted from the simulation trajectories showing convincingly that the isolated C-domain has a similar structure and dynamic behavior than the full arrestin.

We completely agree with the reviewer that this information had been missing. To address this issue, we calculated the average RMSD of simulated C-domains for all simulations MD1 to 4 using as reference structure the corresponding C-domain in the crystallized arrestin (new Supplementary Table S1). The average RMSD values are in all cases below 1.7 Å which convincingly demonstrates that the isolated C-domain remains in a stable conformation compared to the full-length arrestin structure.

This information has been included in the result section on page 4:

“... The conformational stability of the simulated isolated C-domain was steady during all simulation runs with an average RMSD (loops excluded) that did not exceed 1.7 Å compared to the crystallized C-domain of full-length arrestin (Supplementary Table S1).”

Reviewer #1 (Remarks to the Author)

The study of direct arrestin interactions with the membrane and its role in the formation of lower-affinity pre-complex with receptor before the high-affinity binding was greatly improved in revision.

The study is of very high quality. It appropriately combines experiments with molecular simulations. The results are of considerable interest to the scientific community.

Reviewer #3 (Remarks to the Author)

In my opinion, the authors have addressed satisfactorily the concerns by the reviewers. Therefore, I recommend the publication of this manuscript in Nature Communications.

REVIEWERS' COMMENTS:

Reviewer #1 (Remarks to the Author):

The study of direct arrestin interactions with the membrane and its role in the formation of lower-affinity pre-complex with receptor before the high-affinity binding was greatly improved in revision.

The study is of very high quality. It appropriately combines experiments with molecular simulations. The results are of considerable interest to the scientific community.

Reviewer #3 (Remarks to the Author):

In my opinion, the authors have addressed satisfactorily the concerns by the reviewers. Therefore, I recommend the publication of this manuscript in Nature Communications.

We thank the reviewers for their endorsement of our work.